# Multi-Gene Recombinant Baculovirus Expression Systems: From Inception to Contemporary Applications

**DOI:** 10.3390/v16040492

**Published:** 2024-03-23

**Authors:** Sara L. Bissett, Polly Roy

**Affiliations:** Department of Infection Biology, London School of Hygiene and Tropical Medicine, London, WC1E 7HT, UK

**Keywords:** baculovirus expression system, multi-gene transfer vectors, virus assembly, virus-like particle viral vaccines, bluetongue virus

## Abstract

Many protein expression systems are primarily utilised to produce a single, specific recombinant protein. In contrast, most biological processes such as virus assembly rely upon a complex of several interacting proteins rather than the activity of a sole protein. The high complexity of the baculovirus genome, coupled with a multiphase replication cycle incorporating distinct transcriptional steps, made it the ideal system to manipulate for high-level expression of a single, or co-expression of multiple, foreign proteins within a single cell. We have developed and utilised a series of recombinant baculovirus systems to unravel the sequential assembly process of a complex non-enveloped model virus, bluetongue virus (BTV). The high protein yields expressed by the baculovirus system not only facilitated structure–function analysis of each viral protein but were also advantageous to crystallography studies and supported the first atomic-level resolution of a recombinant viral protein, the major BTV capsid protein. Further, the formation of recombinant double-shelled virus-like particles (VLPs) provided insights into the structure–function relationships among the four major structural proteins of the BTV whilst also representing a potential candidate for a viral vaccine. The baculovirus multi-gene expression system facilitated the study of structurally complex viruses (both non-enveloped and enveloped viruses) and heralded a new generation of viral vaccines.

## 1. Introduction

Virus assembly requires the coordinated synthesis and interaction of several gene products. Bluetongue virus (BTV) is a prime example of this architectural complexity, with virions composed of two concentric shells, the outer capsid formed of VP2 and VP5, and the core consisting of VP3 and VP7. Enclosed within the core, the minor viral proteins VP1, VP4 and VP6 reside alongside the viral genome of 10 dsRNA segments. Additionally, BTV encodes three non-structural proteins, NS1, NS2 and NS3 which are synthesised in infected cells [1].

A variety of both prokaryotic and eukaryotic expression systems were developed for the expression of recombinant proteins, and in the field of virology, these technologies were enthusiastically embraced as a means to interrogate the various stages of virus assembly. The baculovirus expression system was adopted for BTV structural studies where it was used to express all 10 viral gene products as individual recombinant proteins [2]. The high protein yields from the baculovirus expression system were advantageous for crystallography studies and facilitated the first atomic structure of a recombinant viral protein, when the major core protein of BTV, VP7, was crystallised as a trimer [3,4].

Each of the 10 BTV proteins is synthesised from a distinct mRNA species with different molar ratios of each protein required for virus assembly, presenting a considerable challenge for BTV structure and function studies [1]. The capacity of the baculovirus genome to incorporate large amounts of foreign DNA and the flexibility of the baculovirus promoters were exploited for the simultaneous expression within a single cell of multiple BTV genes [5]. This process mimicked the various stages of BTV assembly and resulted in the formation of core-like particles (CLPs) from the co-expression of VP3 and VP7, and the formation of BTV virus-like particles (VLPs) from the co-expression of VP2, VP5, VP3 and VP7 [6,7].

This article will review our pioneering work carried out with BTV which established the utility of the baculovirus multi-gene expression system for the study of virus assembly and how the spontaneous formation of VLPs demonstrated the potential of this technology for the production of a new generation of viral vaccines.

### 1.1. Development of Baculovirus Multiple-Gene Transfer Vectors

The ability of the baculovirus system to express two different gene products simultaneously was first demonstrated using the pAcVC2 plasmid [8], which was reconstructed with two copies of the *Autographa californica* nuclear polyhedrosis virus (AcNPV) polyhedron (*ph*) promoter arranged in opposite orientations, paired with appropriate downstream transcription termination sequence. This allowed two distinct genes, the polyhedrin protein of AcNPV and the N protein of lymphocytic choriomeningitis virus (LCMV-N), to be cloned into the pAcVC2 vector under the control of their own copy of the *ph* promoter (Figure 1A). Following co-transfection of insect cells with the pAcVC2 plasmid together with infectious polyhedrin-negative AcNPV DNA, the resulting recombinant baculovirus was shown to express both proteins, which accumulated to high levels in infected insect cells [8].

The pAcVC plasmid was subsequently utilised for the construction of a dual transfer vector which carried the genes encoding the BTV major structural proteins, VP3 and VP7 (Figure 1B) [6]. Analysis of cryoEM images available at the time, and later published [9], suggested that viral cores from authentic BTV virions consisted of two distinct protein layers, with each layer potentially composed of a single protein species. Data available from immunogold staining indicated that VP7 was the principal component of the outermost layer of the core, and suggested that VP3 was likely the other major protein component of the core [10]. A dual-recombinant baculovirus expressing the VP3 and VP7 genes was used to determine whether particles resembling authentic BTV cores could be assembled solely from VP3 and VP7 recombinant proteins. The VP3 and VP7 proteins expressed in insect cells were found to spontaneously interact, forming CLPs which, upon EM analysis, were of the same size and appearance as authentic BTV core particles (Figure 1C) [6]. Dialysis of the CLPs against a low-salt buffer resulted in the loss of their surface capsomeres, revealing a smooth icosahedral subcore which consisted only of VP3, confirming that VP3 forms a scaffold for the assembly of VP7 capsomeres [11]. The formation of recombinant single-shelled core particles demonstrated that VP3 and VP7 were solely responsible for the structural integrity of the core, and that formation was not dependent upon the presence of BTV dsRNA or any of the minor viral proteins which are encapsidated within the core. Additionally, this study also demonstrated that the nonstructural proteins of BTV do not play a facilitatory role in the assembly of empty cores.

The outer capsid of BTV is composed of the major viral proteins VP2 and VP5. VP2 is the antigenic determinant of serotype specificity, harbouring the epitopes that elicit neutralising antibodies [12,13]. Immunisation of sheep with recombinant VP2 protein had been shown to provide a degree of protection from viral challenge [14]. However, this protection was strengthened when a mixture of VP2 and VP5 recombinant proteins were used for immunisation and this coincided with higher neutralising titers detectable in the serum of the sheep. This enhancement of the immune response to the combination of VP2 and VP5 was thought to be due to interactions between the two proteins which resulted in the display of more authentic neutralising epitope conformations, highlighting the importance of VP5 to the structural integrity of VP2.

The simultaneous expression of VP2 and VP5 proteins was undertaken using a novel dual VP2–VP5 recombinant baculovirus; however, no capsid-like structures were detected in infected insect cells [7]. Subsequently, coinfection experiments with both dual recombinant baculoviruses (VP2–VP5 and VP3–VP7) were undertaken to determine whether the core acted as an essential foundation structure for VP2 and VP5 interaction and outer capsid assembly. Coinfection of insect cells resulted in the assembly of empty double-shelled VLPs, with comparable size (~85 nm in diameter) and appearance to authentic BTV particles [7]. It had previously been believed that VP2 and VP5 were present as a ‘diffuse’ protein layer which surrounded the inner viral core [15]; however, the EM analysis of double-shelled BTV VLPs challenged this theory, as an organised arrangement of structural units could be observed on the outer shell of the VLPs. This study represented a major advancement, demonstrating that the assembly of recombinant double-shelled VLPs was possible following the simultaneous co-expression of four major viral structural proteins. Up until this date, only single-shelled VLPs had been successfully expressed [16,17].

Coinfection with both dual recombinant baculovirus successfully produced VLPs; however, the protein preparation also contained a proportion of single-shelled CLPs, due to insect cell infection with the single VP3–VP7 recombinant baculovirus. In order to overcome this obstacle, a quadruple expression vector was constructed which harboured the p10 promoter in duplicate alongside two *ph* promoters (Figure 2A). This facilitated the simultaneous expression of the four structural proteins of BTV from a single recombinant baculovirus, producing protein preparations made up exclusively of VLPs [18].

Multi-gene transfer vectors for the simultaneous expression of two, three, four and five foreign proteins (Figure 2B) were developed to support the study of BTV structure and function [6,18,19]. These vectors primarily utilised the strong *ph* promoter and the *p10* promoter, with foreign gene insertion achieved by either erasing the resident coding sequence or duplicating promoters elsewhere in the baculovirus genome. However, there is a practical limit to the number of genes that can be inserted into a single transfer vector and multiple copies of promoter and termination sequences can be prone to rearrangement and recombination [20]. In order to increase the number of foreign genes which could be simultaneously expressed, an alternative strategy was pursued which identified additional genetic loci (*ctx*, *egt*, *39k*, *orf51*, *gp37*, *iap2* and *odv-e56*) within the baculovirus genome which could support the high-level expression of foreign genes (Figure 3A) [21]. Genomic bacmid DNA was directly modified by red recombination techniques [22] and recombinant viruses were selected using a bipartite selection cassette (Figure 3B). The utility of this system was demonstrated by the co-expression of six genes resulting in the simultaneous formation of influenza A VLPs (M1 and HA) and BTV VLPs (VP2, VP5, VP3 and VP7) [21].

### 1.2. Virus Assembly Studies

BTV CLPs which spontaneously form following infection of insect cells with a dual recombinant baculovirus expressing VP3 and VP7 were subjected to cryo-electron microscopy (cryo-EM) analysis to determine their 3D structure (Figure 4) [23]. The model proposed by this analysis consisted of a spherical subcore-like particle, composed of 60 monomers of VP3 arranged on a T = 1 lattice, and attached to this inner VP3 layer was an icosahedral structure composed of 200 VP7 trimers arranged on a T = 13,l lattice. The recombinant CLPs had a reduced number of VP7 trimers, 200 compared to the 260 which were visualised using authentic virus cores [9]. This was a result of the CLPs lacking five trimers around each of their five-fold axes. There are various explanations for this reduction in VP7 trimers in the recombinant CLPs including the requirement of an additional viral protein to ensure VP7 is securely tethered into place at these five-fold positions [23]. An advantage of using CLPs for the cryo-EM analysis instead of virus cores was the higher contrast which was achieved due to CLPs not possessing an RNA–protein-filled core. This facilitated the visualisation of the, previously unseen, pores in the VP3 layer. These pores were speculated to function by allowing the passage of metabolites and RNA to and from the core for RNA transcription during infection [23].

The 3D reconstruction of recombinant VLPs following cryo-EM analysis revealed an icosahedral structure of 86 nm in diameter, containing all four structural proteins (VP2, VP3, VP5 and VP7) with each protein site highly occupied [24]. This confirmed the spontaneous formation of complete VLPs with essentially the same features as native BTV particles (Figure 4). This study also cleared up an anomaly observed previously, whereby CLPs had a reduced number of VP7 trimers compared to viral cores [23]. The 60 VP7 trimers absent from the CLP were present in the VLP, indicating that the outer capsid proteins, VP2 and VP5, are necessary for the adhesion of these VP7 trimers around the five-fold axes, a finding which agreed with the previously proposed explanation for the absence of these prominent five-fold VP7 trimers in recombinant CLPs.

The multi-gene baculovirus expression system was subsequently utilised to investigate the role played by VP7 in BTV core assembly [26,27]. The outer layer of the core comprises 780 molecules of VP7 which form 260 trimers. These trimers associate into pentameric groupings covering the outer surface of the core, which form intermolecular interactions with the inner VP3 subcore. However, the details underpinning this process were unknown, such as whether the driving force for core assembly was VP7 formation. Was trimer association an ordered process, and how did VP7 trimers attach to the underlying VP3 scaffold? The first study which aimed to address these questions identified key residues within VP7 which were essential for capsid assembly. VP7 proteins were mutated at residues predicted to be involved in monomer–monomer contacts within the trimer and at residues predicted to be in contact with VP3. These mutant VP7 proteins were expressed alongside wildtype (WT) VP3, to assess their impact on VP7 trimer formation, CLP assembly and stability [26]. Of the seven VP7 mutants only one, a substitution of E104W in helix 5, had no apparent effect on stable core formation. The other six mutants, situated in helix 5 (R111F and W119D), helix 6 (F268R and Y271R) and helix 8 (D318N and T321R), demonstrated phenotypic effects, from loss of trimer formation (Y271R) to decreased CLP stability (R111F, W119D and T321R). Furthermore, EM analysis of stable CLPs showed heterogeneous preparations for VP7 mutants F268R and D318N, with the majority of particles in the former made up of solely VP3 subcores [26].

A follow-up study expanded upon these findings and proposed an assembly pathway for the complexed layered core of BTV [27]. Again, the multi-gene baculovirus system was utilised to express mutant VP7 proteins alongside WT VP3, with the resulting protein preparations analysed for trimer formation, CLP assembly and stability. This work demonstrated that the VP7 helix 9 residues G337, P338, A342 and A346 were critical for trimer–trimer interactions during core assembly, since double substitutions at these sites (G337Q/P338Q and A342R/A346R) resulted in poor CLP morphology with only ~60 VP7 molecules retained on the VP3 surface [27]. The effect of these mutations is indirect since helix 9 does not directly interact with the VP3 subcore; instead, they perturb the lateral packing of the VP7 trimers on the VP3 surface. Additionally, it was observed that VP7 mutants with a profound effect on CLP assembly had the capacity to form trimers, indicating that trimerization alone was not sufficient to drive the assembly of the VP7 lattice onto the VP3 core. An alternative model for core assembly was proposed, whereby multiple sheets of VP7 form at different nucleation sites. This process involves the formation of a number of strong VP7 trimer-VP3 contacts which act as multiple equivalent initiations sites, followed by a second set of weaker VP7 trimer interactions which ‘fill in the gaps’ to complete the outer layer of the core (Figure 5) [27].

Once the structural organisation of the BTV core was established, focus turned to better understanding the location of the internal proteins, the viral polymerase VP1 and the viral capping enzyme VP4, which are required for the endogenous transcription activity of the core particles. Cryo-EM analysis of the recombinant CLPs expressed by coinfection of two dual recombinant baculoviruses expressing VP3-VP7 and VP1-VP4 allowed the visualisation of the transcription complex within the core [25]. The 3D reconstruction of the recombinant VP3/7/4/1 CLPs identified a flower-shaped structure attached to the underside of the VP3, directly beneath the fivefold axis, indicating that the VP1 and VP4 likely associate as a heterodimer which directly interacts with VP3. The flower-shaped density was absent from CLPs which contained only one of the viral enzymes, confirming that both proteins needed to be present, and that the formation of a VP1–VP4 complex was essential for occupancy of the VP3 internal fivefold axis position [25]. Such close interaction between the polymerase and the capping enzyme was speculated to be required for the capping of nascent mRNAs as they emerge from the polymerase.

### 1.3. VLPs as Candidate Vaccines for Bluetongue Disease

BTV is transmitted by biting midges of the *Culicoides* species and infects domestic and wild ruminants. Outbreaks of Bluetongue (BT) disease cause significant economic and agricultural burdens with mortality rates reaching as high as 70% in infected sheep [28]. The double-shelled recombinant VLPs expressed by coinfection of two dual recombinant baculoviruses expressing VP2–VP5 and VP3–VP7 were shown to possess potent immunogenic and hemagglutination activity. Antisera raised against VLPs demonstrated high plaque reduction titers (1:10,000) when compared to antisera raised against recombinant VP2 protein alone (<1:640) (Figure 6A) [7], providing further evidence that the authentic presentation of VP2 neutralising epitopes is dependent upon protein–protein interactions with other viral structural proteins [14]. These findings highlighted the potential use of this novel recombinant protein technology for the production of a new generation of viral vaccines which were morphologically and antigenically similar to authentic virions but did not harbour the viral genetic material and replication machinery.

The protective efficacy of VLPs against bluetongue disease was first demonstrated by the immunisation of sheep with recombinant double-shelled VLPs representing BTV10 [29]. Sheep were immunised with different concentrations of VLPs (10–200 µg) adjuvanted with either incomplete Freund’s adjuvant (IFA) or Montanide mannide oleate (ISA-50), with two doses administered 21 days apart. Serum samples taken between days 21 and 117 demonstrated the development of neutralising antibody responses targeting BTV10, which coincided with protection against virulent viral challenge (Figure 6B) [29]. This study demonstrated that immunisation with as little as 10 µg of VLP, the equivalent of ca. 1–2 µg of VP2, could elicit an immune response which was protective against homologous virus challenge. In the context of a VLP, only a small amount of VP2 (1–2 µg) was required to generate a protective immune response compared to the amount required in a crude mixture of recombinant VP2 (50 µg) and VP5 (20 µg) [14]. This superior conformational presentation of neutralising epitopes on the VLP surface highlighted the potential utility of VLPs as a candidate vaccine against BT disease.

A subsequent study expanded the immunogen panel to include VLPs with VP2 proteins representing BTV1, BTV2, BTV13 and BTV17 alongside BTV10. Sheep were immunised with either single or pentavalent VLP formulations before being subjected to homologous (BTV10, BTV13 or BTV17) or heterologous (BTV4, BTV11 or BTV16) virus challenge [30]. In agreement with the previous findings [29], two 10 µg doses of a single VLP formulation were demonstrated to elicit a long-lasting immune response which protected sheep against challenge with the homologous virulent virus. Immunisation with the pentavalent VLP formulation (BTV1/2/10/13/17) resulted in the generation of a polyvalent neutralising antibody response which coincided with protection against homologous virus challenge with BTV13. The results from this study also demonstrated that sheep immunised with 50 µg of BTV10 VLPs were essentially protected from challenge with BTV4, demonstrating the potential for VLPs to generate a cross-protective immune response [30]. However, only partial cross-protection was observed in the sheep immunised with the pentavalent vaccine since half the animals developed mild or no clinical reactions whilst the other half developed moderate clinical reactions following challenge with BTV16. Taken together, these data indicated that immunisation with either single or pentavalent VLP formulations afforded a degree of protection against challenge by heterologous BTV serotypes.

Recombinant VLPs were also utilised to determine the impact of BTV geographical lineage on serotype-specific vaccine efficacy. Molecular epidemiological analysis of European isolates identified that there were two distinct evolutionary lineages, eastern and western, regardless of serotype [31]. Whether intra-serotype geographical lineages differed in their VP2 antigenicity was investigated by immunising sheep with VLPs representing the BTV1 western lineage (isolated in South Africa) prior to virus challenge with BTV1 eastern lineage (isolate in Greece) [32]. All VLP-immunised sheep developed a serum-neutralising antibody response against both the vaccine-incorporated western lineage BTV1 and the non-vaccine-incorporated eastern lineage BTV1. These neutralising antibody responses coincided with protection from challenge with a virulent virus representing the eastern lineage of BTV1, with no clinical manifestations or viremia apparent in sheep post-challenge. The results from this study indicate that the geographical origin of the virus isolate had no significant impact on VP2 antigenicity and was not a critical factor in the development of a serotype-specific candidate vaccine.

Multiple separate outbreaks of BT disease have occurred in Europe over the past two decades, the largest of which was a BTV8 outbreak which occurred between 2006 and 2008 [33]. During the peak of this outbreak, a concurrent BTV1 outbreak emerged, highlighting the necessity for multivalent vaccines. A multivalent vaccine formulation was produced containing recombinant VLPs representing BTV serotypes BTV1, BTV2 and BTV8 to test whether there was immune competition between these genetically diverse serotypes, outbreaks of which cause substantial economic losses [34]. The serum antibodies from the sheep immunised with the multivalent vaccine formulation were able to neutralise all three serotypes in vitro. Additionally, comparable neutralisation titers against BTV8 were observed between the multivalent and a monovalent BTV8 VLP formulation, with both the multivalent and single VLP formulations being highly efficacious at preventing viraemia or the clinical manifestation of BT disease after challenge with BTV8. This study demonstrated that a strong, protective immune response was developed following immunisation with VLP representing different BTV serotypes with no observable immune competition [34].

Recombinant VLP vaccines for BTV are not limited by concerns associated with attenuated and killed vaccines, and more importantly, allow differentiation between infected and vaccinated animals (DIVA). Certain issues are inherent with the use of attenuated and killed vaccines such as batch-to-batch variation of virus inactivation. However, specific issues regarding the use of these vaccines are apparent in cattle, including the development of mild BT symptoms [35] and decreased fertility [36]. Additionally, attenuated and killed vaccines are not DIVA compliant and vaccinated cattle are subject to movement and trade restrictions. In contrast, VLPs are safe and efficacious immunogens that are able to afford complete protection against virulent virus challenge and can also be rapidly produced in large quantities to respond to outbreak situations.

### 1.4. VLPs Representing Complex Enveloped Viruses

The outbreak of severe acute respiratory syndrome coronavirus 1 (SARS-CoV-1) in 2003 was the first pandemic of the 21st Century; however, this was dwarfed in comparison with the worldwide pandemic which began in 2020, following the outbreak of severe acute respiratory syndrome coronavirus 2 (SARS-CoV-2) [37]. Both pandemic viruses belong to the family *Coronaviridae* and have a positive-sense, single-stranded RNA genome surrounded by a viral envelope composed of a host cell-derived lipid bilayer in which the viral structural proteins spike (S), envelope (E) and membrane (M) are anchored [38]. The expression of these three membrane-associated proteins, S, E and M, and their assembly into enveloped coronavirus VLPs represented a challenge for heterologous expression systems.

The appeal of VLPs, which can be generated rapidly and at high concentrations, as vaccine candidates for emerging diseases drove further innovations in the recombinant protein field, with SARS-CoV-1 VLPs successfully produced using the multi-gene recombinant baculovirus expression system [39]. This study demonstrated that the co-expression of the coronavirus membrane-associated proteins S, E and M, at very high levels from a triple recombinant baculovirus (Figure 7A), allowed the spontaneous assembly of highly stable VLPs, with glycosylated S protein forming very distinct spikes on the surface of the VLPs (Figure 7B). Assembled VLPs budded from the plasma membrane of the infected insect cells acquiring a lipid bilayer, mimicking the release of authentic virions from the host cell. Mice immunised with SARS-CoV-1 VLPs generated serum antibody responses which, in vitro, were able to neutralise a retrovirus, pseudotyped with the spike protein of SARS-CoV-1, at comparable titres to those observed in serum taken from a SARS-CoV-1 convalescent patient (Figure 7C) [40]. These data provided evidence that the SARS-CoV-1 VLPs were immunogenic and that the epitope conformations on the recombinant spike protein were antigenically similar to authentic spike epitopes.

Recombinant SARS-CoV-2 VLPs representing the original Wuhan isolate were produced in the same manner, following the expression of the three viral structural proteins S, E and M from a single recombinant baculovirus (Figure 8A). In agreement with the findings for SARS-CoV-1 VLPs, SARS-CoV-2 VLPs demonstrated the relevant antigenic characteristics and immunogenic properties [41]. In addition, the Syrian hamster challenge model was utilised to determine the protective efficacy of the SARS-CoV-2 VLPs [42]. Two groups of hamsters were immunised with either two doses of non-adjuvanted VLPs or buffer alone prior to intranasal challenge with the Alpha variant (B.1.1.7), with infection monitored by recovery of virus from oral and nasopharyngeal washes. The hamsters were also assessed for weight loss and clinical signs of lung pathology following virus challenge. Immunisation with SARS-CoV-2 VLPs, in the absence of adjuvant, reduced virus shedding following heterologous challenge with a virulent virus and protected hamsters from disease-associated weight loss (Figure 8B). Examination of lung tissue from both the VLP immunised and control groups demonstrated differential pathology, particularly for markers of inflammation, where the VLP group showed a reduced overall score at 10 days post-challenge [41]. Cumulatively, these data demonstrated that in a challenge model SARS-CoV-2, VLPs were immunogenic in the absence of adjuvant, mitigated viral load and prevented the development of severe disease associated with SARS-CoV-2 infection.

## 2. Conclusions

The development of multiple-gene transfer vectors and the subsequent generation of recombinant baculoviruses were pivotal in harnessing the full potential of recombinant protein technology. The simultaneous expression of multiple foreign proteins within a single cell, which could spontaneously self-assemble to form particles morphologically similar to viral cores and whole viruses, offered a technique which could be exploited for the purpose of virus assembly studies. Subsequent studies provided novel insights into the steps of BTV assembly and the structural organisation of this complex double-layered virus. The multi-gene recombinant baculovirus expression system facilitated the expression of VLPs representing complex enveloped and non-enveloped viruses, which demonstrated immunogenicity and antigenicity similar to authentic virions and highlighted the potential use of recombinant VLPs as the basis for novel viral vaccine candidates. This innovative work led the way and established solid foundations for future studies across the field of virology.

## Figures and Tables

**Figure 1 viruses-16-00492-f001:**
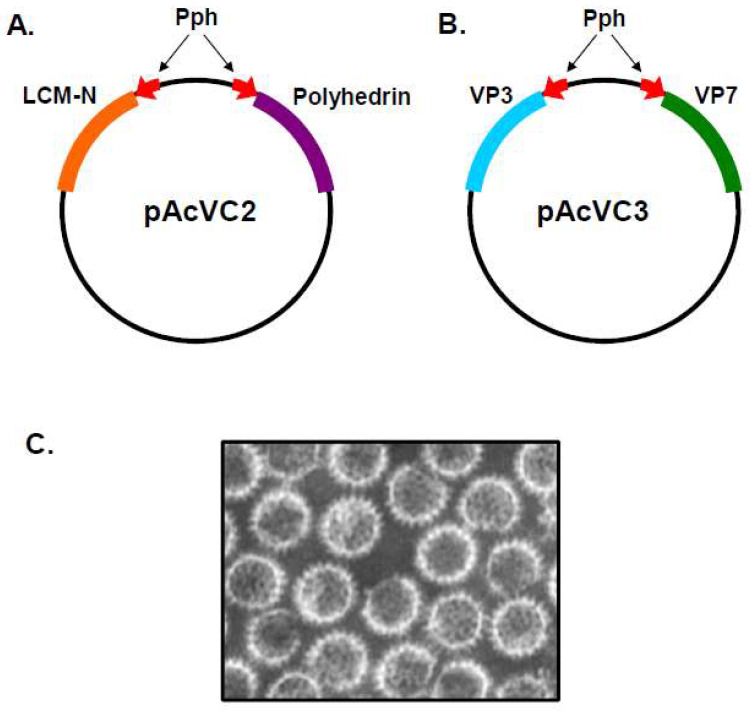
Baculovirus Dual Gene Transfer Vector for CLP Expression. General organisation of vectors pAcVC2 and pAcVC3. Red arrows represent polyhedron promoter (Pph) and indicate the direction of transcription. Viral genes are represented by coloured blocks. (**A**) pAvCV2 contains genes expressing the LCMV-N and polyhedrin proteins. (Adapted from [8]). (**B**) In pAvCV3, the genes expressing LCMV-N and polyhedrin have been replaced with genes expressing the VP7 and VP3 of BTV (adapted from [6]). (**C**) EM of empty BTV CLPs comprised of the two major core proteins VP3 and VP7, expressed from a dual recombinant baculovirus (Adapted from [7]).

**Figure 2 viruses-16-00492-f002:**
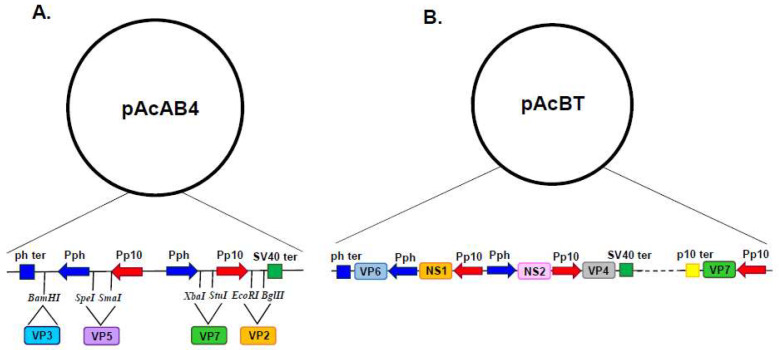
Baculovirus Quadruple and Quintuple Gene Transfer Vector. General organisation of vectors pAcAB4 and pAcBT. Arrows represent promoters and indicate the direction of transcription. Blue arrows represent polyhedron promoter (Pph) and red arrows represent p10 promoter (Pp10). Boxes represent transcriptional termination signals with the polyhedron (ph) termination signal in blue, the SV40 termination signal in green and the p10 termination signal in yellow. (**A**) In pAcAB4, the restriction sites relative to the promoter and termination signals are indicated along with the four BTV genes inserted into the pAcAB4 vector (Adapted from [18]). (**B**) In pAcBT, the five BTV genes inserted into the pAcBT vector are indicated, adjacent to their relevant promoters and termination signals (Adapted from [19]).

**Figure 3 viruses-16-00492-f003:**
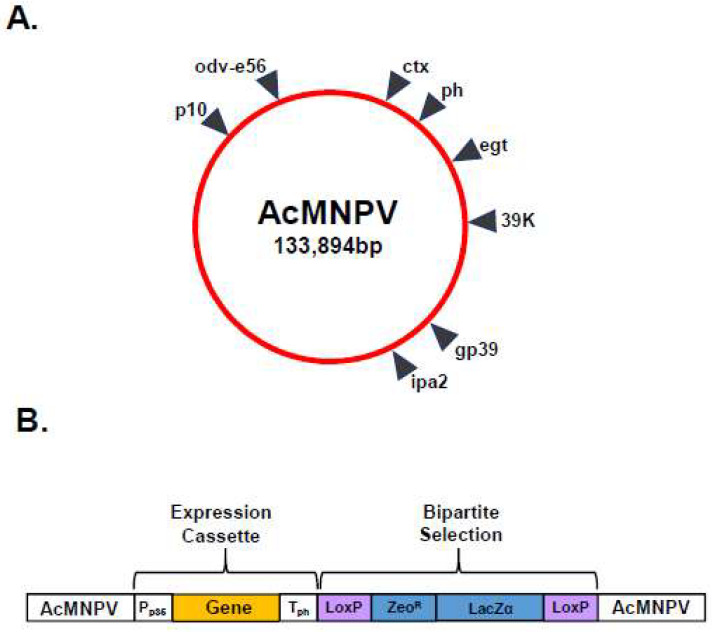
Modification of Genomic Bacmid DNA for Multi-gene Expression. (**A**) Diagram of AcMNPV genome showing additional genetic loci identified within the baculovirus genome which could support high-level expression of foreign genes. (**B**) Recombinant baculoviruses were selected using a bipartite selection cassette comprising LoxP sites (purple boxes) flanking a Zeocin resistance gene (Zeo^R^) and LacZα marker (blue boxes). Foreign genes (orange box) were inserted between the p35 promoter (P_p35_) and the polyhedrin polyadenylation sequences (T_ph_) (white boxes) (adapted from [21]).

**Figure 4 viruses-16-00492-f004:**
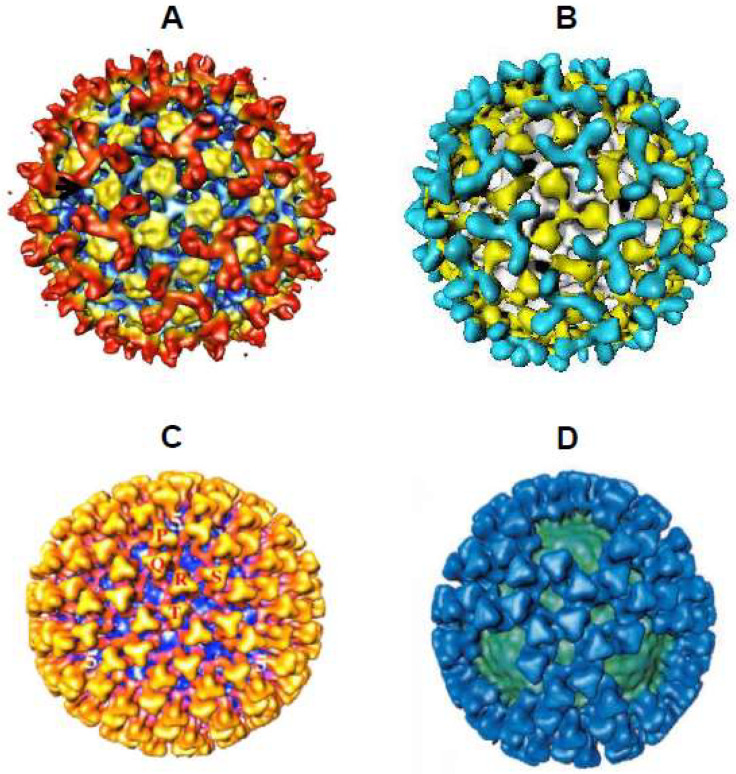
Comparison of 3D Cryo-EM Structures. Comparison of authentic virus and core structures with recombinant double-shelled VLPs and single-shelled CLPs. Recombinant proteins form structures which mimic the BTV core and virion particles. The BTV VP2 trimers are represented on the virion (**A**) and VLP (**B**) structures in red or cyan, respectively, and the VP5 trimers are yellow in both images. The VP7 trimers are represented on the core (**C**) and CLP (**D**) in yellow or blue, respectively, and the VP3 layer is coloured blue or green, respectively (Adapted from [24,25]).

**Figure 5 viruses-16-00492-f005:**
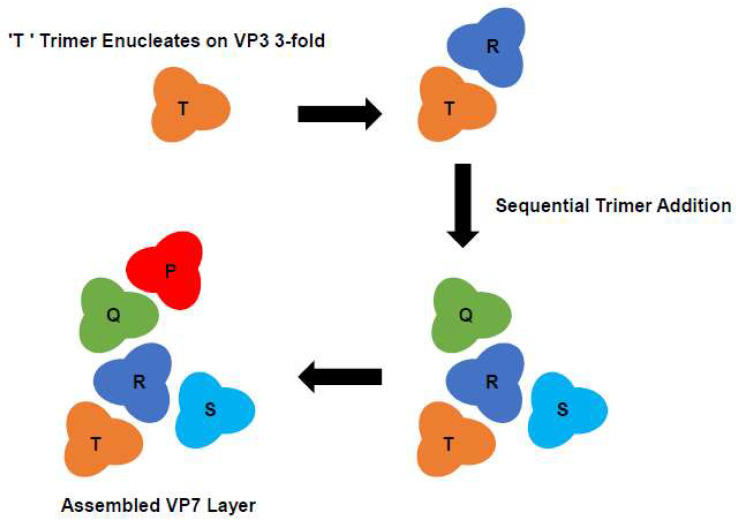
Diagram of Proposed VP7 Assembly Pathway. VP7 trimers are labelled P, Q, R, S and T. In the proposed model of core assembly, multiple sheets of VP7 form at different nucleation sites. A process which is initiated by a single VP7 trimer (‘T’) occupying the ‘preferred’ site for assembly on the VP3. The ‘T’ trimer becomes the most tightly attached trimer, which is subsequently followed by the progressively weaker attachment of four more VP7 trimers (R-Q-S-P) which ‘fill the gaps’ (Adapted from [27]).

**Figure 6 viruses-16-00492-f006:**
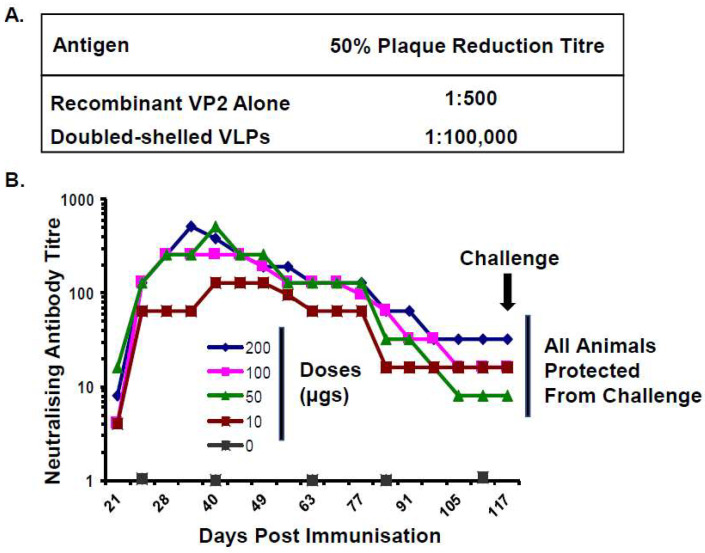
Serum Neutralising Antibody Responses Following Immunisation with Recombinant VP2 or Different Doses of Double-Shelled VLPs. (**A**) Plaque reduction serum antibody titres against BTV10 following immunisation with recombinant VP2 alone or doubled-shelled VLPs, demonstrating the difference in 50% titres. (Adapted from [7]) (**B**) Average serum neutralising antibody titres from sheep (*n* = 4) immunised with two doses of VLPs at 10, 50, 100 or 200 µg. At day 117, sheep were challenged with virulent BTV. No signs of BT disease or viremia were detected in any of the VLP-immunised animals (Adapted from [29]).

**Figure 7 viruses-16-00492-f007:**
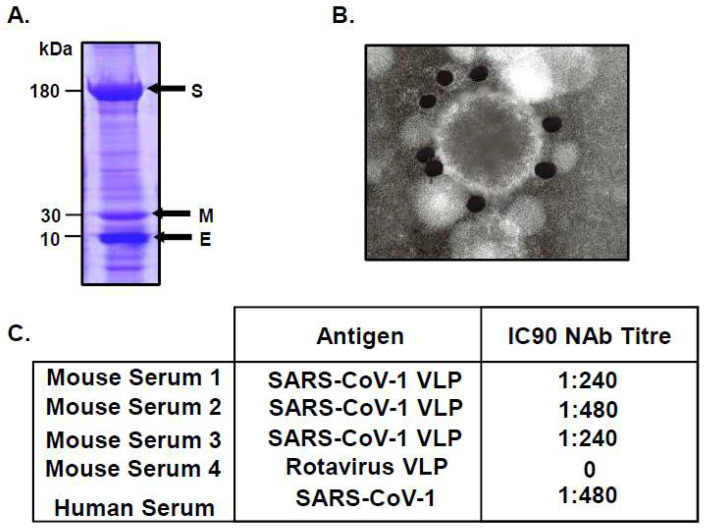
SARS-CoV-1 VLP Characterisation and Neutralising Antibody Responses. (**A**) Expression of S, M and E SARS-CoV-1 proteins in insect cells infected with a triple recombinant baculovirus, analysed by SDS-PAGE followed by Coomassie blue staining. (**B**) EM of immunogold-labelled SARS-CoV-1 VLPs probed with anti-S monoclonal antibody (adapted from [39]). (**C**) IC90 neutralising antibody titre against SARS-CoV-1 pseudotyped lentivirus, using sera from 3 mice immunised with SARS-CoV-1 VLPs, 1 mouse immunised with Rotavirus VLP and one serum obtained from a SARS-CoV-1 convalescent patient (adapted from [40]).

**Figure 8 viruses-16-00492-f008:**
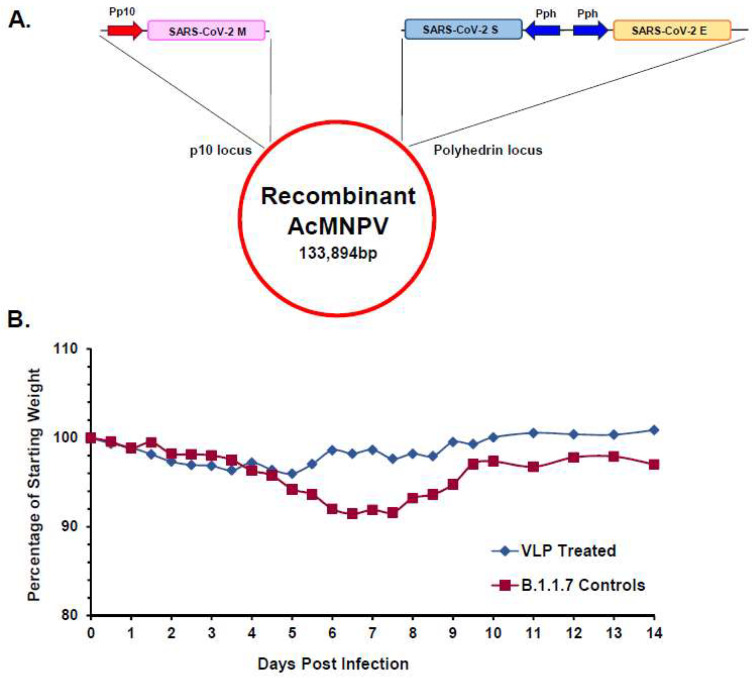
Design of Multi-gene Transfer Vector for SARS-CoV-2 VLP Expression and Efficacy of SARS-CoV-2 VLPs against Disease Associated Weight Loss following Heterologous Challenge. (**A**) Diagram of gene arrangement in the transfer vector used to generate the recombinant baculovirus expressing SARS-CoV-2 VLPs. The M protein was expressed from the p10 locus whilst the S and E proteins were expressed by back-to-back polyhedrin promoters (ph) within the ph locus. (**B**) Average weight loss of Syrian hamster groups following live virus challenge. Closed green circles represent VLP-immunised animals. Closed red circles represent the control animals (adapted from [41]).

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
