# Peer review of "Multi-Gene Recombinant Baculovirus Expression Systems: From Inception to Contemporary Applications"

_viruses, 2024, doi:10.3390/v16040492_

Round 1

Reviewer 1 Report

Comments and Suggestions for Authors

In this review manuscript, the authors comprehensively discussed the multi-gene recombinant baculovirus expression systems which allowed the simultaneous expression of multiple virus like particle (VLP) proteins within a single cell and the spontaneous assembly of virus like cores or particles morphologically. In addition, the authors highlighted the important application of this technology for the structural dissection of BTV-core/VLP organizations as well as VLP-based novel viral vaccine development. The recently emerged technologies of single gene transcript-derived expression of multiple genes in a single mammalian cell or multi-gene transfer baculovirus vector-derived expression of multiple genes in a single insect cell have been well used to product virus like particles (VLP), both non-enveloped VLPS and enveloped-VLPS, for the safe and efficacious vaccine development. This review is interesting and useful to the readers in the fields. The rationale and objective are clear and fully addressed in the review manuscript

The following minors may be useful to improve the quality of the review manuscript:

1.     Since one of the major contexts associated with the multi-gene recombinant baculovirus expression systems used to VLP expression or production, it is better for the authors to alter the tittle to address this improvement. 

2.     This review manuscript is well-written, but there are several occurrences such as some meaningful confusion and expressing errors that require a proof prior to publication. For instance: 

(1). In the lines 303 – 305, “However, only partially cross-protection was observed in the sheep immunised with the pentavalent vaccine, were challenged with BTV16 resulted in half the animals developing mild or no clinical reactions whilst the other half developed moderate clinical reactions.”

(2) In the lines 362 – 363, “This study demonstrated that the co-expression of the coronavirus membrane-associated proteins S, E and M, at very high levels from a single recombinant baculovirus (Figure 7A), allowed the spontaneous assembly of highly stable VLPs,”, but in the lines 375 – 376 (Figure 7A), “(A) Expression of S, M and E SARS-CoV-1 proteins in insect cells infected with a triple recombinant baculovirus,”

(3). In the lines 411 - 413, “The development of multiple-gene transfer vectors and the subsequent generation of recombinant baculoviruses, where pivotal in harnessing the full potential of recombinant protein technology.”

Comments on the Quality of English Language

This review manuscript is well-written, but there are several occurrences such as some meaningful confusion and expressing errors that require a proof prior to publication.

Author Response

We thank the reviewer for their comments and suggestions. We have addressed all the points highlighted for minor revisions. Please find our responses below.

  1. Since one of the major contexts associated with the multi-gene recombinant baculovirus expression systems used to VLP expression or production, it is better for the authors to alter the tittle to address this improvement. 

Authors Response: Proposed new title ‘Multi-gene recombinant baculovirus expression systems: From inception to contemporary applications’

  1. This review manuscript is well-written, but there are several occurrences such as some meaningful confusion and expressing errors that require a proof prior to publication. For instance: 

(1). In the lines 303 – 305, “However, only partially cross-protection was observed in the sheep immunised with the pentavalent vaccine, were challenged with BTV16 resulted in half the animals developing mild or no clinical reactions whilst the other half developed moderate clinical reactions.”

Authors Response: This sentence has been restructured: ‘However, only partially cross-protection was observed in the sheep immunised with the pentavalent vaccine since half the animals developed mild or no clinical reactions whilst the other half developed moderate clinical reactions following challenge with BTV16.’ Please see lines 304 – 306.

(2) In the lines 362 – 363, “This study demonstrated that the co-expression of the coronavirus membrane-associated proteins S, E and M, at very high levels from a single recombinant baculovirus (Figure 7A), allowed the spontaneous assembly of highly stable VLPs,”, but in the lines 375 – 376 (Figure 7A), “(A) Expression of S, M and E SARS-CoV-1 proteins in insect cells infected with a triple recombinant baculovirus,”

Authors Response: The word ‘single’ has been replaced with the word ‘triple’. Please see line 365. 

(3). In the lines 411 - 413, “The development of multiple-gene transfer vectors and the subsequent generation of recombinant baculoviruses, where pivotal in harnessing the full potential of recombinant protein technology.”

Authors Response: The word ‘where’ has been replaced with the word ‘were’. Please see line 413.

Reviewer 2 Report

Comments and Suggestions for Authors

Manuscript ID: viruses-2890278

 “Virus-like particles as viral vaccines: From inception to contemporary applications”

 This is a review article focused on the use of the baculovirus expression system for the generation and characterization Bluetongue virus (BTV) virus-like particles (VLPs). It reviews the extensive work carried out by the authors developing the baculovirus multigene expression system, for the study of BTV virus assembly, the generation of VLPs and their use as new generation viral vaccines. I think the paper is clearly written and condenses relevant information on the subject. I only have minor comments.

 Minor comments:

Title: The title is quite generic (virus-like particles as viral vaccines), while the review is strongly focused on BTV VLPs (with a brief part dedicated to SARS-CoV VLPs) and their generation using the baculovirus expression system. Maybe this could be reflected in the title.

 Lines 138-139: “Genomic bacmid DNA was directly modified by red recombination techniques” I think it would be useful to provide a brief explanation (and or reference) of “red recombination techniques” as it may not be obvious for everyone.

 Lines 140-142: “The utility of this system was demonstrated by the co-expression of six genes resulting in the simultaneous formation of influenza A VLPs (M1 & HA) and BTV VLPs (VP2, VP5, VP3 & VP7)”. A reference here would be useful.

 Line 166-167: “The recombinant CLPs had a reduced number of VP7 copies, 200” This sentence is misleading because it is “200 VP7 trimers”, not “200 VP7 copies.”

 Lines 310-312. “Molecular epidemiological analysis of European isolates had identified that there were two distinct evolutionary lineages, eastern and western, regardless of serotype.” A reference here would be useful.

 Line 319: “the western lineage of BTV1”. I think this is incorrect. It seems the correct is “eastern lineage” as indicated in line 315. Please check.

 There are some small errors or wrong expressions like:

 - Line 54 (at the end): “2”

 - Line 56: “Results” In a review article there is no Results section.

 Line 63: “LCMV-N” and in lines 100-106 (legend to figure 1) “LCM-N”. It would be better to use the same acronym (maybe “LCMV-N”).

 Line 75: “[Hyatt_1988]” a reference in wrong format.

 - Line 115 “which surrounding the inner viral core”

- Line 241 “needed to be presence”

 - Lines 303-304: “However, only partially cross-protection was observed in the sheep immunised with the pentavalent vaccine, were challenged with BTV16 resulted in half the animals developing…” This sentence is awkward.

Author Response

We thank the reviewer for their comments and suggestions. We have addressed all the points highlighted for minor revisions. Please find our responses below.

Title: The title is quite generic (virus-like particles as viral vaccines), while the review is strongly focused on BTV VLPs (with a brief part dedicated to SARS-CoV VLPs) and their generation using the baculovirus expression system. Maybe this could be reflected in the title.

Authors Response: Proposed new title ‘Multi-gene recombinant baculovirus expression systems: From inception to contemporary applications’

Lines 138-139: “Genomic bacmid DNA was directly modified by red recombination techniques” I think it would be useful to provide a brief explanation (and or reference) of “red recombination techniques” as it may not be obvious for everyone.

Authors Response: Cited Datsenko, K.A. and B.L. Wanner, One-step inactivation of chromosomal genes in Escherichia coli K-12 using PCR products. Pro Natl Acad Sci USA, 2000. 97(12) p6640-45. Please see line 139, Reference [22].

Lines 140-142: “The utility of this system was demonstrated by the co-expression of six genes resulting in the simultaneous formation of influenza A VLPs (M1 & HA) and BTV VLPs (VP2, VP5, VP3 & VP7)”. A reference here would be useful.

Authors Response: Cited Noad, R.J., et al., Multigene expression of protein complexes by iterative modification of genomic Bacmid DNA. BMC Mol Biol, 2009. 10: p. 87. Please see line 143, Reference [21].

Line 166-167: “The recombinant CLPs had a reduced number of VP7 copies, 200” This sentence is misleading because it is “200 VP7 trimers”, not “200 VP7 copies.”

Authors Response: Have replaced the word ‘copies’, with the word ‘trimers’ in line 168.

Lines 310-312: “Molecular epidemiological analysis of European isolates had identified that there were two distinct evolutionary lineages, eastern and western, regardless of serotype.” A reference here would be useful.

Authors Response: Cited Maan, S., et al., Molecular epidemiology studies of bluetongue virus. 2009: Academic Press. Please see line 313, Reference [31].

Line 319: “the western lineage of BTV1”. I think this is incorrect. It seems the correct is “eastern lineage” as indicated in line 315. Please check.

Authors Response: Have replaced the word ‘western’, with the word ‘eastern’ in line 320.

Line 54: (at the end): “2”

Authors Response: “2” deleted from the end of the sentence.

Line 56: “Results” In a review article there is no Results section.

Authors Response: “Results” heading has been deleted.

Line 63: “LCMV-N” and in lines 100-106 (legend to figure 1) “LCM-N”. It would be better to use the same acronym (maybe “LCMV-N”).

Authors Response: In lines 103 & 104, ‘LCM-N’ has been changed to ‘LCMV-N’.

Line 75: “[Hyatt_1988]” a reference in wrong format.

Authors Response: Changed to correct reference format.

Line 115: “which surrounding the inner viral core”

Authors Response: Changed ‘surrounding’ to ‘surrounded’.

Line 242 “needed to be presence”

Authors Response: Changed ‘presence’ to ‘present’.

Lines 303-304: “However, only partially cross-protection was observed in the sheep immunised with the pentavalent vaccine, were challenged with BTV16 resulted in half the animals developing…” This sentence is awkward.

Authors Response: Sentence has been restructured: ‘However, only partially cross-protection was observed in the sheep immunised with the pentavalent vaccine since half the animals developed mild or no clinical reactions whilst the other half developed moderate clinical reactions following challenge with BTV16.’